# Estimating weighted areas under the ROC curve

**Andreas Maurer**
Istituto Italiano di Tecnologia
am@andreas-maurer.eu

**Massimiliano Pontil**
Istituto Italiano di Tecnologia & University College London
massimiliano.pontil@iit.it

## Abstract

Exponential bounds on the estimation error are given for the plug-in estimator of weighted areas under the ROC curve. The bounds hold for single score functions and uniformly over classes of functions, whose complexity can be controlled by Gaussian or Rademacher averages. The results justify learning algorithms which select score functions to maximize the empirical partial area under the curve (pAUC). They also illustrate the use of some recent advances in the theory of nonlinear empirical processes.

## 1 Introduction

Using the area under the ROC curve (the AUC) to evaluate score functions has a long history in medical screening and bioinformatics ([8],[9]). In the last decades several algorithms have been developed to learn score functions which maximize the AUC ([6], [24]). In some cases, however, different regions of the false positive range may not be equally relevant to assess the quality of the score and should therefore be weighted differently, say with some nonconstant weight function $W$. In melanoma detection, for example, false negatives have disastrous consequences for a patient, while a certain number of false positives is tolerable. A good scoring function should then have very large values on the right hand side of the ROC plot. Such considerations have created interest in a *partial* area under the ROC curve (the pAUC), which only measures the area between two specified false positive rates, so that $W$ is a step function ([26], [15], [18]). Several algorithms have been designed with the goal of maximizing the pAUC over classes of scoring functions ([20], [21], [19], [22]).

Any measurement or optimization of the AUC or pAUC must rely on a finite number of observations, which raises the questions of estimation and generalization. These problems are well understood for the AUC ([2], [5], [23]), where $W$ is constant, but for more general weight functions estimation may be difficult to impossible. In Proposition 1 below we will show that the bias of its plug-in estimator may be bounded away from zero even in the limit of an infinite sample.

Our first contribution shows that these problems are absent if the weight function $W$ is Lipschitz continuous. Theorem 2 shows that then the weighted AUC can be estimated at rate $n^{-1/2}$ by its plug-in estimator, where $n$ is the sample size. Theorem 3 shows that the Lipschitz-weighted AUC can be uniformly estimated over a class of score functions at rate $n^{-1/2}$ if the Gaussian complexity of the class increases as $n^{1/2}$, which is standard for most function classes in machine learning. This result also implies a statistical performance guarantee for algorithms maximizing the empirical pAUC as in [21] or [19].

Even if $W$ is Lipschitz the weighted AUC remains a nonlinear statistical functional of challenging complexity. Our second contribution is Theorem 7, which provides a general method to control the estimation errors of any statistical functional, which satisfies certain Lipschitz conditions with respect to the total-variation, Wasserstein or Kolmogorov metrics. The verification of these conditions for the weighted AUC then provides the proofs of Theorem 2 and Theorem 3.

## 1.1 Related work

The pAUC estimator is a special case of an L-estimator (see [25]). Asymptotic results for L-estimators have been known for a long time. Helmers [10] gave Berry-Esseen type rates of normal approximation for smoothened L-statistics. Finite sample bounds for special distributions are shown in [3]. The asymptotics of different estimators for the pAUC are considered in [7]. More recent asymptotic results for the pAUC are given in [27] and [1]. [13] announces a uniform bound of order $n^{-1/4}$ for left-partial areas. We do not know of any finite sample bounds comparable to Theorem 2 or uniform bounds of order $n^{-1/2}$ comparable to Theorem 3 below.

## 1.2 Notation

With $\mathbf{1}_S$ we denote the indicator function of a set $S$ and with $H := \mathbf{1}_{(0,\infty)}$ the Heaviside function. $\mathcal{P}(\mathcal{X})$ is the set of non-negative measures and $\mathcal{P}_1(\mathcal{X}) \subset \mathcal{P}(\mathcal{X})$ is the set of probability measures on a measurable space $\mathcal{X}$ respectively. For $\lambda \in [0,1]$ and $\mu, \nu \in \mathcal{P}(\mathcal{X})$ we write $\lambda(\mu, \nu)$ for the convex combination $\lambda(\mu, \nu) = \lambda\mu + (1-\lambda)\nu$. If $x \in \mathcal{X}$ then $\delta_x \in \mathcal{P}_1(\mathcal{X})$ is the unit mass at $x$. If $\mu \in \mathcal{P}_1(\mathcal{X})$ and $\mathbf{X} = (X_1, ..., X_n) \sim \mu^n$ is an iid sample of size $n$ the empirical measure is $\hat{\mu}(\mathbf{X}) = \frac{1}{n}\sum_{i=1}^{n}\delta_{X_i} \in \mathcal{P}_1(\mathcal{X})$. If $\mathbf{X}$ is unambiguous we write $\hat{\mu} = \hat{\mu}(\mathbf{X})$.

$\mathcal{Y}^{\mathcal{X}}$ is the set of functions $f : \mathcal{X} \to \mathcal{Y}$, for $f \in \mathcal{Y}^{\mathcal{X}}$ and $\mu \in \mathcal{P}(\mathcal{X})$ the push-forward $f_{\#}\mu \in \mathcal{P}(\mathcal{Y})$ is defined $f_{\#}\mu(A) = \mu(f^{-1}(A))$, $A \subseteq \mathcal{Y}$. For $f \in \mathbb{R}^{\mathcal{X}}$ $\|f\|_{\infty} = \sup_{x \in \mathcal{X}}|f(x)|$ and if $(\mathcal{X}, d)$ is a metric space then $\|f\|_{Lip} = \sup_{x \neq y \in \mathcal{X}}|f(x) - f(y)|/d(x,y)$.

For $\mu \in \mathcal{P}_1(\mathcal{X} \times \{0,1\})$ (random labeled data) $\mu_0$ and $\mu_1 \in \mathcal{P}(\mathcal{X})$ are defined $\mu_0(A) = \mu(A \times \{0\})$ and $\mu_1(A) = \mu(A \times \{1\})$ for $A \subseteq \mathcal{X}$, and $\mu(0) = \mu(\mathcal{X} \times \{0\})$ and $\mu(1) = \mu(\mathcal{X} \times \{1\})$ denote the relative frequencies of the two labels.

A summary of notation in tabular form is given in the supplement.

## 2 Weighted areas under the ROC-curve

Underlying the concept of the ROC-curve is the joint random occurrence of scores $X$ in some open, bounded interval $\mathcal{I} \subseteq \mathbb{R}$ and binary labels $Y \in \{0,1\}$. We assume $(X, Y) \sim \mu$ for some law $\mu \in \mathcal{P}_1(\mathcal{I} \times \{0,1\})$, with corresponding un-normalized measures $\mu_0, \mu_1 \in \mathcal{P}(\mathcal{I})$ and scalar frequencies $\mu(0)$ and $\mu(1)$.

Any threshold $t \in \mathcal{I}$ on a score $x$ induces a second labeling $H(x - t)$ alternative to $y$. The second labeling is considered correct iff it coincides with $y$. Correspondingly the true positive rate and false positive rate are the functions $tpr, fpr : \mathcal{I} \to [0,1]$ defined respectively as

$$tpr(t) = \frac{\mu_1(t, \infty)}{\mu(1)} \text{ and } fpr(t) = \frac{\mu_0(t, \infty)}{\mu(0)}.$$

If we assume that $\mu_0$ has a positive density, then $fpr$ has a unique inverse, and the ROC-curve can be defined as the function $roc : [0,1] \to [0,1]$

$$roc(u) = tpr\left(fpr^{-1}(u)\right).$$

The ROC-curve gives the true positive rate corresponding to a specified false positive rate. Knowledge of the ROC-curve allows us to adjust the classification threshold according to respective cost estimates for false positives and false negatives This accounts for the great importance of ROC-curves in practice.

Let $W : [0,1] \to \mathbb{R}_+$ be some specified weight function and consider the quantity

$$\int_{[0,1]} roc(u)W(1-u)\,du.$$

With the change of variables $u = fpr(t)$ this is seen to be equal to

$$f_{W,\ell}(\mu) = \frac{1}{\mu(1)\mu(0)}\int_{\mathcal{I}^2}\ell(t' - t)W\left(\frac{\mu_0(-\infty, t]}{\mu(0)}\right)d\mu_1(t')\,d\mu_0(t), \tag{1}$$

as long as $\ell$ is the Heaviside function, $\ell = H$ (see supplement). Other choices of $\ell : \mathbb{R} \to [0, 1]$ replacing or approximating $H$ will play a role when we want to learn scoring functions to optimize properties of the ROC-curve, but $\ell$ will generally be assumed nondecreasing. The equation (1) makes sense even when $fpr$ is not uniquely invertible, and it will serve as our definition of the statistical functional $f_{W,\ell} : \mathcal{P}_1 (\mathcal{I} \times \{0, 1\}) \to \mathbb{R}_+$ in the sequel. Note that $f_{W,\ell}$ is monotonic in $W$ and $\ell$, and that its range lies in $[0, \|W\|_\infty]$.

An important special case is the "area under the curve" (AUC) obtained if $W \equiv 1$.

$$f_{1,H} (\mu) = \int_{[0,1]} roc (u) \, du = \frac{1}{\mu (1) \mu (0)} \int_{\mathcal{I}^2} H (t' - t) \, d\mu_1 (t') \, d\mu_0 (t).$$

$f_{1,H} (\mu)$ is the probability that $t' > t$ if $t'$ and $t$ are drawn independently from the positive and the negative conditional distributions. The AUC has been widely used to measure the quality of score functions and its statistical properties have been thoroughly analyzed from different perspectives ([6], [2], [5]).

If the respective costs of false negatives and false positives are very different, different regions of the false positive range are not equally relevant to measure the quality of the score. This leads to the consideration of a weighted or partial AUC where $W$ is nonconstant and assigns different weights to different false positive rates. In the simplest case $W$ it is a step function, such as $\mathbf{1}_{[a,b]}$ with $0 \le a < b \le 1$, but $W$ may also be chosen to approximate the value of the ROC-curve itself at a given point.

## 2.1 Estimation

The measure $\mu$, and with it the ROC-curve itself, are mathematical idealizations, accessible only through observations. We assume that we have access to an iid sample $(\mathbf{X}, \mathbf{Y}) = ((X_1, Y_1), ..., (X_n, Y_n)) \sim \mu^n$ of labeled data. This defines the empirical measure $\hat{\mu} = \hat{\mu} (\mathbf{X}, \mathbf{Y}) \in \mathcal{P}_1 (\mathcal{I} \times \{0, 1\})$ and corresponding empirical variants $\hat{\mu}_0, \hat{\mu}_1$ as well as the empirical rates $\hat{\mu} (0)$ and $\hat{\mu} (1)$. We will study the estimation of $f_{W,\ell} (\mu)$ by the plug-in estimator

$$
\begin{aligned}
f_{W,\ell} (\hat{\mu}) &= \frac{1}{\hat{\mu} (1) \hat{\mu} (0)} \int_{\mathcal{I}^2} \ell (t' - t) W \left( \frac{\hat{\mu}_0 (-\infty, t]}{\hat{\mu} (0)} \right) d\hat{\mu}_1 (t') \, d\hat{\mu}_0 (t) \\
&= \frac{1}{|S_1| |S_0|} \sum_{i \in S_1} \sum_{j \in S_0} \ell (X_i - X_j) W \left( \frac{|\{k \in S_0 : X_k \le X_j\}|}{|S_0|} \right),
\end{aligned}
$$

where $S_1 = \{i : Y_i = 1\}$ and $S_0 = \{i : Y_i = 0\}$.

The dependence of the weight function on the order statistic of $\hat{\mu}_0 / \hat{\mu} (0)$ causes problems which are absent in the estimation of the AUC as in [2]. If $W$ has a discontinuity then the value of the functional depends discontinuously on the underlying law in the weak topology, and random fluctuations can make estimation impossible, even if the factor $1 / (\hat{\mu} (1) \hat{\mu} (0))$ is bounded. The supplement gives proof of the following proposition, which shows that for discontinuous $W$ the bias of the plug-in estimator may be bounded away from zero even in the limit of infinite sample sizes.

**Proposition 1** *For $W = \mathbf{1}_{[1/2,1]}$ there exists a bounded, open interval $\mathcal{I}$ and $\mu \in \mathcal{P}_1 (\mathcal{I} \times \{0, 1\})$ such that for every even $n$ and $\mathbf{X} \sim \mu^n$*

$$\lim_{n \to \infty} \mathbb{E}_{\mathbf{X} \sim \mu^n} [f_{W,H} (\hat{\mu} (\mathbf{X}))] = 1/4 < 1/2 = f_{W,H} (\mu).$$

To resolve this problem one might assume some regularity of $\mu$, but since we wish to estimate a property of an *unknown* distribution from observations, we prefer to assume that the weight function $W$ is Lipschitz.

We still need to exclude the case that one of the observed label frequencies $\hat{\mu} (0)$ and $\hat{\mu} (1)$ is too small relative to the sample size. For $\delta > 0$ we define the event

$$A_\delta = \left\{ \min \{\hat{\mu} (0), \hat{\mu} (1)\} > \sqrt{\frac{2 \ln (4/\delta)}{n}} \right\}. \tag{2}$$

We then have the following result.

**Theorem 2** *Suppose* $W : \mathbb{R} \to [0, \infty)$ *satisfies* $\|W\|_\infty$, $\|W\|_{Lip} < \infty$, *that* $\ell : \mathcal{I} \to [0, 1]$ *is nondecreasing and* $\mu \in \mathcal{P}_1 (\mathcal{I} \times \{0, 1\})$. *Then for any* $\delta > 0$ *we have with probability at least* $1 - \delta$ *in* $(\mathbf{X}, \mathbf{Y}) \sim \mu^n$ *that* $A_\delta$ *implies*

$$\left| f_{W, \ell} (\hat{\mu} (\mathbf{X}, \mathbf{Y})) - f_{W, \ell} (\mu) \right| \leq \frac{\|W\|_{Lip} + 9 \|W\|_\infty}{\min \{\hat{\mu} (0), \hat{\mu} (1)\}^2} \sqrt{\frac{2 \ln (4/\delta)}{n}}.$$

**Remarks:**

1. For any event $B$ the statement *"with probability at least* $1 - \delta$ *it holds that* $A_\delta$ *implies* $B$" means $\Pr \{A_\delta \cap B^c\} < \delta$, where $B^c$ is the complement of $B$, so if we observe $A_\delta$ the bound holds with high probability.

2. Since $f_{W, H}$ is monotonic in $W$ the bound implies one-sided bounds for discontinuous weight functions, such as the step functions defining the pAUC. If $\hat{W} \leq W_{Lip} \leq W$ where $\hat{W}$ and $W$ are discontinuous and $W_{Lip}$ is Lipschitz, then with high probability $f_{W, H} (\mu) \geq f_{\hat{W}, H} (\hat{\mu}) - O(1/\sqrt{n})$. Upper estimates and sandwich estimates may be constructed similarly.

3. As in the previous remark an approximation to the value of $roc$ at some $u \in [0, 1]$ can be estimated with plug-in estimators $f_{\hat{W}_n, H} (\hat{\mu})$, if the weight functions $\hat{W}_n$ are Lipschitz and, if regarded as probability densities, converge weakly to the unit mass $\delta_{1-u}$.

## 2.2 Uniform bounds

Now let $\mathcal{X}$ be some space of instances and let $\mu \in \mathcal{P}_1 (\mathcal{X} \times \{0, 1\})$ be a law for labeled instances. Suppose that $\mathcal{H} \subseteq \mathcal{I}^{\mathcal{X}}$ is a class of candidate scoring functions $h : \mathcal{X} \to \mathcal{I}$. For every $h \in \mathcal{H}$ define $\bar{h} : \mathcal{X} \times \{0, 1\} \to \mathcal{I} \times \{0, 1\}$ by $\bar{h} (x, y) = (h(x), y)$. Given a sample $(\mathbf{X}, \mathbf{Y}) \sim \mu^n$, we would like to find $h \in \mathcal{H}$ so as to (approximately) maximize the value of $f_{W, H} (\bar{h}_\# \mu)$, where $\bar{h}_\# \mu$ is push-forward of $\mu$ under $\bar{h}$.

The strategy is to pick an $h$ which (approximately) maximizes a regularized empirical surrogate $f_{W', \ell} (\bar{h}_\# \hat{\mu} (\mathbf{X}, \mathbf{Y}))$, where $W'$ and $\ell$ are Lipschitz lower bounds of $W$ and $H$ respectively. The situation mirrors that of support vector machines, where one minimizes the hinge-loss as Lipschitz upper bound of the 0-1-loss.

Since the chosen score function $h$ depends on $(\mathbf{X}, \mathbf{Y})$, it becomes a random variable, and a justification of the method is given by a high probability bound on

$$\sup_{h \in \mathcal{H}} f_{W', \ell} (\bar{h}_\# \hat{\mu}) - f_{W, H} (\bar{h}_\# \mu) \leq \sup_{h \in \mathcal{H}} f_{W', \ell} (\bar{h}_\# \hat{\mu}) - f_{W', \ell} (\bar{h}_\# \mu),$$

where the inequality follows from $f_{W', \ell} \leq f_{W, H}$. We can bound the right hand side.

**Theorem 3** *Suppose* $W : \mathbb{R} \to [0, \infty)$ *satisfies* $\|W\|_\infty$, $\|W\|_{Lip} < \infty$, *that* $\ell : \mathcal{I} \to [0, 1]$ *is nondecreasing and* $\|\ell\|_{Lip} < \infty$, *that* $\mathcal{H} \subseteq \mathcal{I}^{\mathcal{X}}$ *and that* $\mu \in \mathcal{P}_1 (\mathcal{X} \times \{0, 1\})$. *Let* $A_\delta$ *be as in* (2). *Then for any* $\delta > 0$ *we have with probability at least* $1 - \delta$ *in* $(\mathbf{X}, \mathbf{Y}) \sim \mu^n$ *that* $A_\delta$ *implies*

$$\sup_{h \in \mathcal{H}} \left| f_{W, \ell} (\bar{h}_\# \mu) - f_{W, \ell} (\bar{h}_\# \hat{\mu}) \right|$$

$$\leq \frac{8 \sqrt{2 \pi} \|\ell\|_{Lip} \left( \|W\|_\infty + \|W\|_{Lip} \right)}{\hat{\mu} (0)^2 \hat{\mu} (1)} \frac{G(\mathcal{H})}{n} + \frac{\|W\|_{Lip} + 10 \|W\|_\infty}{\min \{\hat{\mu} (0), \hat{\mu} (1)\}^2} \sqrt{\frac{4 \ln (16 n/\delta)}{n}}.$$

*where* $G(\mathcal{H})$ *is the expected Gaussian complexity* $G(\mathcal{H}) = \mathbb{E}_{\mathbf{X}} \mathbb{E}_\gamma \sup_{h \in \mathcal{H}} \sum_{i=1}^{n} \gamma_i h(X_i)$, *with independent standard normal variables* $\gamma_1, ..., \gamma_n$.

**Remarks.**

1. Again we must observe $A_\delta$ for the bound to hold with high probability.

2. The use of Rademacher and Gaussian complexities has become a standard in learning theory, and there is a large body of literature giving bounds ([4], [14], [12], [11]), which can be substituted

above. These bounds apply to many different function classes, such as multi-layer neural networks or bounded sets of linear functionals in a reproducing-kernel-Hilbert space. Typically the bounds on the Gaussian complexity are of $O\left(\sqrt{n \ln (n)}\right)$ making the above bound $O\left(\sqrt{\ln (n)/n}\right)$.

3. Existing algorithms of [21] or [19] optimize the pAUC for discontinuous weight functions $\hat{W}$ and practitioners may wish guarantees for discontinuous weight functions $W$. The next corollary, which directly follows from the monotonicity of $f_{W,\ell}$ in $W$ and $\ell$, shows that we can give such guarantees of $O\left(\sqrt{\ln (n)/n}\right)$, whenever a Lipschitz weight function $W_{Lip}$ can be jammed between $\hat{W}$ and $W$. This can be regarded as a statistical guarantee for the algorithms of [21] and [19].

**Corollary 4** *Let* $\hat{W}, W_{Lip}, W : [0,1] \to [0,\infty)$, $\hat{W} \leq W_{Lip} \leq W$ *and* $\|W_{Lip}\|_{Lip} < \infty$. *Then with probability at least* $1 - \delta$ *in* $(\mathbf{X}, \mathbf{Y}) \sim \mu^n$ *we have that* $A_\delta$ *implies*

$$\forall h \in \mathcal{H}, \ f_{W,H}\left(\left(\bar{h}_{\#}\mu\right)\right) \geq f_{\hat{W},\ell}\left(\bar{h}_{\#}\hat{\mu}\right) - B\left(n, W_{Lip}, \mathcal{H}, \hat{\mu}, \delta\right),$$

*where* $B\left(n, W_{Lip}, \mathcal{H}, \hat{\mu}, \delta\right)$ *is the bound in Theorem 3.*

## 3 Proofs

In this section we outline the proofs. Several technical details are given in the supplementary material.

### 3.1 Conditioning on the empirical label frequencies

The appearance of the true label frequencies $\mu(0)$ and $\mu(1)$ in the various denominators in the definition (1) of $f_{W,\ell}$ is a nuisance. But if the weight function is Lipschitz, we can approximate the estimation difference for $f_{W,\ell}$ by the estimation difference of another functional $g_{W,\ell,c}$, which is independent of the label frequencies and defined for $c > 0$ as

$$g_{W,\ell,c}(\mu) := \int_{\mathcal{I}} \int_{\mathcal{I}} \ell\left(t' - t\right) d\mu_1\left(t'\right) W\left(\frac{\mu_0\left(-\infty, t\right]}{c}\right) d\mu_0(t).$$

In situations, where $W, \ell$ and $c$ are unambiguously fixed, we will simply write $g$ for $g_{W,\ell,c}$.

**Lemma 5** *Let* $\delta \in (0,1)$ *and* $A_\delta$ *as in (2). Then with probability at least* $1 - \delta/2$ *it holds that* $A_\delta$ *implies both* $\mu(0) \geq \hat{\mu}(0)/2$ *and*

$$\left|f_{W,\ell}(\hat{\mu}) - f_{W,\ell}(\mu)\right| \leq \frac{\left|g_{W,\ell,\mu(0)}(\hat{\mu}) - g_{W,\ell,\mu(0)}(\mu)\right|}{\hat{\mu}(0)\,\hat{\mu}(1)} + \frac{\|W\|_{Lip} + 8\,\|W\|_\infty}{\min\{\hat{\mu}(0),\hat{\mu}(1)\}^2}\sqrt{\frac{2\ln(4/\delta)}{n}}.$$

The proof, a straightforward application of Hoeffdings inequality, is given in the supplement. If $W$ is Lipschitz and $\hat{\mu}(0)$ and $\hat{\mu}(1)$ and $n$ are reasonably large, the lemma allows us to bound the estimation error for $f_{W,\ell}$ in terms of the estimation error of the functional $g_{W,\ell,c}$, for which we have the following result.

**Proposition 6** *(i) For* $\delta > 0$ *with probability at least* $1 - \delta$

$$\left|g_{W,\ell,c}(\mu) - g_{W,\ell,c}(\hat{\mu})\right| \leq \|W\|_\infty \sqrt{\frac{2\ln(2/\delta)}{n}}. \tag{3}$$

*(ii) Under the conditions of Theorem 3 we have with probability at least* $1 - \delta$ *in* $(\mathbf{X}, \mathbf{Y}) \sim \mu^n$ *that*

$$\sup_{h \in \mathcal{H}} \left|\mathbb{E}g_{W,\ell,c}\left(\bar{h}_{\#}\hat{\mu}\right) - g_{W,\ell,c}\left(\bar{h}_{\#}\hat{\mu}\left(\mathbf{X},\mathbf{Y}\right)\right)\right|$$

$$\leq \frac{4\sqrt{2\pi}\,\|\ell\|_{Lip}\left(2\,\|W\|_\infty + c^{-1}\,\|W\|_{Lip}\right)}{n}G(\mathcal{H}) + 2\,\|W\|_\infty \sqrt{\frac{\ln(2/\delta)}{n}}.$$

The proof of this proposition is not easy, but it easily implies Theorem 2 and Theorem 3.

**Proof.** (Theorem 2) A union bound of the inequalities in (i) and Lemma 5 together with some algebraic simplifications. ∎

**Proof.** (Theorem 3) Set $\delta = n^{-1/2}$ in (3) and take the expectation inside the absolute value. Using $0 \leq g_{W,\ell,c} \leq \|W\|_\infty$ this gives for every $\mu$ (and every $\bar{h}_{\#}\mu$) the bias bound

$$|g_{W,\ell,c}(\mu) - \mathbb{E}g_{W,\ell,c}(\hat{\mu})| \leq \|W\|_\infty \left( \sqrt{\frac{2\ln(2\sqrt{n})}{n}} + \sqrt{\frac{1}{n}} \right) \leq 2\|W\|_\infty \sqrt{\frac{\ln(4n)}{n}}.$$

Combine this with (ii), set $c = \mu(0)$ and simplify to obtain

$$\sup_{h \in \mathcal{H}} \left| g_{W,\ell,\mu(0)}(\bar{h}_{\#}\mu) - g_{W,\ell,\mu(0)}(\bar{h}_{\#}\hat{\mu}(\mathbf{X}, \mathbf{Y})) \right|$$

$$\leq \frac{4\sqrt{2\pi} \|\ell\|_{Lip} \left( 2\|W\|_\infty + \mu(0)^{-1} \|W\|_{Lip} \right)}{n} G(\mathcal{H}) + 2\|W\|_\infty \sqrt{\frac{2\ln(8n/\delta)}{n}}.$$

Now assume $A_\delta$ and combine with Lemma 5 in a union bound, using also $\mu(0) \geq \hat{\mu}(0)/2$, and simplify to obtain the conclusion. ∎

### 3.2 Plug-in estimators for Lipschitz functionals

To establish Proposition 6 we will give a general method to prove high probability bounds on the estimation and uniform estimation error for Lipschitz functionals on $\mathcal{P}_1(\mathcal{U})$ with $\mathcal{U} \subseteq \mathbb{R}$.

For any $\mu, \nu \in \mathcal{P}(\mathbb{R})$ we define the metrics

$$
\begin{aligned}
d_{TV}(\mu, \nu) &= |\mu - \nu|(\mathbb{R}) \\
d_1(\mu, \nu) &= \int_{\mathbb{R}} |\mu((-\infty, t]) - \nu((-\infty, t])| \, dt \\
d_\infty(\mu, \nu) &= \sup_{t \in \mathbb{R}} |\mu((-\infty, t]) - \nu((-\infty, t])|.
\end{aligned}
$$

If $\mu, \nu \in \mathcal{P}_1(\mathbb{R})$ then $d_{TV}$ is the total variation metric, $d_1$ is the 1-Wasserstein metric and $d_\infty$ is the Kolmogorov metric. All these metrics have the convexity property

$$d(\lambda(\mu, \nu), \lambda(\mu', \nu')) \leq \lambda d(\mu, \mu') + (1 - \lambda) d(\nu, \nu') \tag{4}$$

for any $\lambda \in [0, 1]$ and $\mu, \mu', \nu, \nu' \in \mathcal{P}_1(\mathbb{R})$.

**Theorem 7** *Let $\mathcal{U} \subseteq \mathbb{R}$ and $f : \mathcal{P}_1(\mathcal{U}) \to \mathbb{R}$ and let $\delta > 0$. Consider the following conditions on $f$*

*(a) $\forall \mu, \nu \in \mathcal{P}_1(\mathcal{U})$, $f(\mu) - f(\nu) \leq L_\infty d_\infty(\mu, \nu)$.*

*(b) $\forall \mu, \nu \in \mathcal{P}_1(\mathcal{U})$, $f(\mu) - f(\nu) \leq L_1 d_1(\mu, \nu)$.*

*(c) $\forall \lambda \in [0, 1]$ and $\forall \mu, \mu', \nu, \nu' \in \mathcal{P}_1(\mathcal{U})$ we have*

$$g(\lambda(\mu, \nu)) - g(\lambda(\mu, \nu')) - g(\lambda(\mu', \nu)) + g(\lambda(\mu', \nu')) \leq (1 - \lambda)\lambda L_2 d_1(\nu, \nu') d_{TV}(\mu, \mu').$$

*Then*

*(i) Suppose $f$ satisfies (a). Let $\mu \in \mathcal{P}_1(\mathcal{U})$ and $\hat{\mu} = \hat{\mu}(\mathbf{X})$ with $\mathbf{X} \sim \mu^n$. Then with probability at least $1 - \delta$*

$$|f(\mu) - f(\hat{\mu})| \leq L_\infty \sqrt{\frac{\ln(2/\delta)}{2n}}.$$

*(ii) Suppose $f$ satisfies (a),(b) and (c). Let $\mathcal{H} \subseteq \mathcal{U}^\mathcal{X}$, $\mu \in \mathcal{P}_1(\mathcal{X})$ and $\hat{\mu} = \hat{\mu}(\mathbf{X})$ with $\mathbf{X} \sim \mu^n$. Then with probability at least $1 - \delta$*

$$\sup_{h \in \mathcal{H}} f(h_{\#}\hat{\mu}) - \mathbb{E}f(h_{\#}\hat{\mu}) \leq \frac{\sqrt{8\pi}(L_1 + L_2)}{n} G(\mathcal{H}) + L_\infty \sqrt{\frac{\ln(1/\delta)}{n}}.$$

The proof uses two nontrivial auxiliary results. For (i) we need the following version of the Dvoretzky-Kiefer-Wolfowitz theorem as sharpened by Massart [16].

**Theorem 8** *If $\mu$ is a probability measure on the real line and $\hat{\mu}$ is the empirical measure for $\mathbf{X} \sim \mu^n$ then for $t > 0$*

$$\Pr\{d_\infty(\mu, \hat{\mu}) > t\} \leq 2e^{-2nt^2}.$$

For part (ii) we use a result about nonlinear empirical processes [17], for which we need some additional notation. For $\mathbf{x} \in \mathcal{X}^n$ and $h \in \mathcal{U}^{\mathcal{X}}$ we write $h(\mathbf{x}) = (h(x_1), ..., h(x_n)) \in \mathcal{U}^n$. For $k \in \{1, ..., n\}$ and $y, y' \in \mathcal{U}$ we define the partial difference operator for functions $f : \mathcal{U}^n \to \mathbb{R}$ by

$$D_{y,y'}^k f(\mathbf{x}) = f(x_1, ..., x_{k-1}, y, x_k, ..., x_n) - f(x_1, ..., x_{k-1}, y', x_k, ..., x_n).$$

**Theorem 9** *(see [17]) Let $\mathbf{X} = (X_1, ..., X_n)$ be a vector of independent random variables with values $\mathcal{X}$ and $\mathbf{X}'$ iid to $\mathbf{X}$. Let $\mathcal{U} \subseteq \mathbb{R}$, $\mathcal{H} \subseteq \mathcal{U}^{\mathcal{X}}$ and $f : \mathcal{U}^n \to \mathbb{R}$. Then for any $\delta \in (0, 1)$, with probability at least $1 - \delta$,*

$$\sup_{h \in \mathcal{H}} f(h(\mathbf{X})) - \mathbb{E}[f(h(\mathbf{X}'))] \leq \sqrt{2\pi}(2M_L(f) + J_L(f))\, G(\mathcal{H}) + M(f)\sqrt{n\ln(1/\delta)},$$

*where the three seminorms $M$, $J_L$ and $M_L$ are defined as*

$$M(f) = \max_k \sup_{\mathbf{x} \in \mathcal{U}^n, y, y' \in \mathcal{U}} D_{y,y'}^k f(\mathbf{x})$$

$$M_L(f) = \max_k \sup_{\mathbf{x} \in \mathcal{U}^n, y, y' \in \mathcal{U}, y \neq y'} \frac{D_{y,y'}^k f(\mathbf{x})}{|y - y'|}$$

$$J_L = n \max_{k,l:k \neq l} \sup_{\mathbf{x} \in \mathcal{U}^n, z, z', y, y' \in \mathcal{U}, y \neq y'} \frac{D_{y,y'}^k D_{z,z'}^l f(\mathbf{x})}{|y - y'|}.$$

**Proof.** (of Theorem 7) (i) If (a) holds then by Theorem 8

$$\Pr\{|g(\mu) - g(\hat{\mu})| > t\} \leq \Pr\left\{d_\infty(\mu, \hat{\mu}) > \frac{t}{L_\infty}\right\} \leq 2\exp\left(\frac{-2nt^2}{L_\infty^2}\right).$$

Set the right hand side to $\delta$ and solve $t$ for the conclusion.

(ii) Suppose (a),(b) and (c) hold. We will use Theorem 9 and bound the seminorms for $f(\hat{\mu})$. Fix $\mathbf{x} \in \mathcal{U}^n$ and $k \in \{1, ..., n\}$ and set $\hat{\mu}_k = \frac{1}{n-1}\sum_{i:i \neq k}\delta_{x_k}$.

$$D_{y,y'}^k f(\hat{\mu}) = f\left(\frac{n-1}{n}\hat{\mu}_k + \frac{1}{n}\delta_y\right) - f\left(\frac{n-1}{n}\hat{\mu}_k + \frac{1}{n}\delta_{y'}\right)$$

$$\leq L_\infty d_\infty\left(\frac{n-1}{n}\hat{\mu}_k + \frac{1}{n}\delta_y, \frac{n-1}{n}\hat{\mu}_k + \frac{1}{n}\delta_{y'}\right) \leq \frac{L_\infty}{n}d_\infty(\delta_y, \delta_{y'}) \leq \frac{L_\infty}{n},$$

where the first inequality follows from (a), the second from convexity $d_\infty$ (see (4)) and the last from $d_\infty(\delta_y, \delta_{y'}) \leq 1$. In the same way (b) gives $D_{y,y'}^k f(\hat{\mu}) \leq L_1 |y - y'|/n$, since $d_1(\delta_y, \delta_{y'}) = |y - y'|$. This gives $M(f) \leq L_\infty/n$ and $M_L(f) \leq L_1/n$.

To bound $J_L(f)$ let $l \neq k$ and write $\hat{\mu}_{kl} = \frac{1}{n-2}\sum_{i:i \notin \{k,l\}}\delta_{x_k}$, $\hat{\mu}_{k,z} = \frac{n-2}{n-1}\hat{\mu}_{kl} + \frac{1}{n-1}\delta_z$ and $\hat{\mu}_{k,z'} = \frac{n-2}{n-1}\hat{\mu}_{kl} + \frac{1}{n-1}\delta_{z'}$. Then

$$D_{y,y'}^k D_{z,z'}^l g(\mathbf{x})$$

$$= f\left(\frac{n-1}{n}\hat{\mu}_{k,z} + \frac{1}{n}\delta_y\right) - f\left(\frac{n-1}{n}\hat{\mu}_{k,z} + \frac{1}{n}\delta_{y'}\right)$$

$$\quad - f\left(\frac{n-1}{n}\hat{\mu}_{k,z'} + \frac{1}{n}\delta_y\right) + f\left(\frac{n-1}{n}\hat{\mu}_{k,z'} + \frac{1}{n}\delta_{y'}\right)$$

$$\leq \frac{n-1}{n^2}L_{12}d_{TV}(\hat{\mu}_{k,z}, \hat{\mu}_{k,z'})\,d_1(y, y') \leq \frac{L_2}{n^2}d_{TV}(\delta_z, \delta_{z'})\,d_1(y, y') \leq \frac{2L_2}{n^2}|y - y'|.$$

The first inequality follows from (c), the second from convexity of $d_{TV}$ (see (4)) and the last from $d_{TV}(\delta_z, \delta_{z'}) \leq 2$. This gives $J_L(f) \leq 2L_2/n$. Substitution in the conclusion of Theorem 9 completes the proof. ∎

### 3.3 Proof of Proposition 6

To apply Theorem 7 we need to control the Lipschitz properties of $g_{W,\ell,c}$. We do so at first with respect to the unnormalized measures $\mu_0$ and $\mu_1$.

**Proposition 10** *Fix $W, \ell$ and $c$. The functional $g = g_{W,\ell,c}$ satisfies $\forall \mu, \mu', \nu, \nu' \in \mathcal{P}_1 \left( \mathcal{I} \times \{0,1\} \right)$*

*(a) $g(\nu) - g(\nu') \leq \|W\|_\infty \left( d_\infty (\nu_0, \nu'_0) + d_\infty (\nu_1, \nu'_1) \right)$*

*(b) if $\ell$ is Lipschitz then $g(\nu) - g(\nu') \leq \|W\|_\infty \|\ell\|_{Lip} \left( d_1 (\nu_0, \nu'_0) + d_1 (\nu_1, \nu'_1) \right)$*

*(c) if $W$ and $\ell$ are Lipschitz then*

$$g\left(\lambda\left(\mu,\nu\right)\right) - g\left(\lambda\left(\mu,\nu'\right)\right) - g\left(\lambda\left(\mu',\nu\right)\right) + g\left(\lambda\left(\mu',\nu'\right)\right)$$
$$\leq (1-\lambda)\,\lambda\,\|\ell\|_{Lip}\left(\frac{1}{c}\|W\|_{Lip} + \|W\|_\infty\right)\left(d_1\left(\nu_1,\nu'_1\right) + d_1\left(\nu_0,\nu'_0\right)\right)d_{TV}\left(\mu,\mu'\right).$$

The proof is given in the supplement. We then eliminate the labels and replace the disconnected space $\mathcal{I} \times \{0,1\}$ by a subset of the real line. Let $a = \sup \mathcal{I}$ and $b = \sup \mathcal{I} - \inf \mathcal{I}$. We map $\mathcal{I} \times \{0\}$ to $(-b, 0)$ in ascending order of $\mathcal{I}$, and we map $\mathcal{I} \times \{1\}$ to $(0, b)$ in descending order of $\mathcal{I}$.

More formally we define the bijection $\tau : (x,y) \in \mathcal{I} \times \{0,1\} \mapsto (2y-1)(a-x) \in (-b,0) \cup (0,b)$. The push-forward $\mu \mapsto \tau_\# \mu$ is then a bijection between $\mathcal{P}_1 \left( \mathcal{I} \times \{0,1\} \right)$ and $\mathcal{P}_1 \left( (-b,0) \cup (0,b) \right)$ with inverse $\tau_\#^{-1}$. Then for $A \subset \mathcal{I}$ we have $\mu_0(A) = \mu(A \times \{0\}) = \tau_\# \mu (A - a)$ and $\mu_1(A) = \tau_\# \mu (a - A)$. The supplement gives proof of the following lemma.

**Lemma 11** *For $\mu, \nu \in \mathcal{P}_1 \left( \mathcal{I} \times \{0,1\} \right)$ we have*

*(i) $d_{TV} (\mu_0, \nu_0) + d_{TV} (\mu_1, \nu_1) = d_{TV} (\mu, \nu) = d_{TV} (\tau_\# \mu, \tau_\# \nu)$*

*(ii) $d_\infty (\mu_0, \nu_0) + d_\infty (\mu_1, \nu_1) \leq 2 d_\infty (\tau_\# \mu, \tau_\# \nu)$*

*(iii) $d_1 (\mu_0, \nu_0) + d_1 (\mu_1, \nu_1) \leq 2 d_1 (\tau_\# \mu, \tau_\# \nu)$*

Since $g_{W,\ell,c}(\mu) = g_{W,\ell,c} \circ \tau_\#^{-1} (\tau_\# \mu)$, we can shift perspective to the functional $g_{W,\ell,c} \circ \tau_\#^{-1}$ on $\mathcal{P}_1 \left( (-b,0) \cup (0,b) \right)$. Proposition 10 and Lemma 11 show that the functional $\mu \mapsto g_{W,\ell,c} \circ \tau_\#^{-1} (\mu)$ satisfies the Lipschitz condition (a) of Theorem 7 with $L_\infty = 2 \|W\|_\infty d_\infty (\mu, \nu)$ and, if $\ell$ is Lipschitz, also (b) with $L_1 = \|W\|_\infty \|\ell\|_{Lip}$ and (c) with

$$L_2 = 2 \|\ell\|_{Lip} \left( \frac{1}{c} \|W\|_{Lip} + \|W\|_\infty \right).$$

In Theorem 7 we then replace $\mathcal{U}$ by $(-b,0) \cup (0,b)$, $f$ by $g_{W,\ell,c} \circ \tau_\#^{-1}$, $\mathcal{X}$ by $(\mathcal{X} \times \{0,1\})$, $\mathcal{H}$ by the class of functions $\mathcal{H}' = \{(x,y) \mapsto \tau(h(x), y) : h \in \mathcal{H}\}$, and we substitute the constants $L_\infty, L_1$ and $L_2$ by the values given above. Theorem 7 then gives us parts (i) and (ii) of Proposition 6, since by symmetry of the standard normal distribution

$$G(\mathcal{H}') = \mathbb{E}_\mathbf{X} \mathbb{E}_\gamma \sup_{h \in \mathcal{H}} \sum_{i=1}^{n} \gamma_i (2Y_i - 1)(a - h(X_i)) = G(\mathcal{H}).$$

## 4  Conclusion

In a nonparametric setting the estimation of partial areas under the ROC-curve by plug-in estimators was shown to be impossible. It is nevertheless possible to give error bounds for Lipschitz weight functions and uniform bounds which provide some justification for existing algorithms optimizing the pAUC. The method to control the uniform estimation errors for nonlinear statistical functionals in terms of Lipschitz conditions appears promising in a more general context.

## Acknowledgments and Disclosure of Funding

This work was partially supported by a grant from SAP SE.

## Broader Impact

A solid mathematical basis is beneficial to the development of practical statistical methods. We believe that the present work improves the understanding of ROC-curves and the optimization of score functions used in machine learning and medical diagnostics.

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
