[Supplementary Material]

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

# Appendix

For the convenience of the reader we restate results which appear in the main part of the paper.

## A  Proof of the formula (1) for the pAUC

**Lemma 12** *We have*

$$\int_{[0,1]} roc\,(u)\,W\,(1-u)\,du = f_{W,H}\,(\mu)\,.$$

**Proof.** With $u = fpr\,(t) = \frac{\mu_0(t,\infty)}{\mu(0)}$ we get

$$
\begin{aligned}
&\int_{[0,1]} roc\,(u)\,W\,(1-u)\,du \\
&= \int_{[0,1]} tpr\,\left(fpr^{-1}\,(u)\right)\,W\,(1-u)\,du \\
&= \frac{1}{\mu\,(0)} \int_{\mathbb{R}} tpr\,(t)\,W\,\left(\frac{\mu_0\,(-\infty,t]}{\mu\,(0)}\right)\,d\mu_0\,(t) \\
&= \frac{1}{\mu\,(0)\,\mu\,(1)} \int_{\mathbb{R}} \int_{\mathbb{R}} H\,(t'-t)\,d\mu_1\,(t')\,W\,\left(\frac{\mu\,((-\infty,t],0)}{\mu\,(0)}\right)\,d\mu_0\,(t) \\
&= f_{W,H}\,(\mu)\,,
\end{aligned}
$$

since

$$tpr\,(t) = \frac{\mu_1\,(t,\infty)}{\mu\,(1)} = \frac{1}{\mu\,(1)} \int_{\mathbb{R}} H\,(t'-t)\,d\mu_1\,(t')\,.$$

■

## B  Proof of the lower bound on the bias, Proposition 1

**Proposition 1** *For $W = 1_{[1/2,1]}$ there exists a bounded, open interval $I$ and $\mu \in P_1\,(\mathcal{I} \times \{0,1\})$ such that for every even $n$ and $X \sim \mu^n$*

$$\lim_{n \to \infty} \mathbb{E}_{\mathbf{X} \sim \mu^n}\,[f_{W,H}\,(\hat{\mu}\,(\mathbf{X}))] = 1/4 < 1/2 = f_{W,H}\,(\mu)\,.$$

**Proof.** (of Proposition 1) Let $\mathcal{I}$ be any bounded open interval containing $[0,1]$ and consider members of $\mathcal{P}_1\,(\mathcal{I} \times \{0,1\})$ supported on the set $\{(0,0)\,,(1/2,1)\,,(1,0)\}$. Every such probability measure is of the form

$$\mu_{s,r} := s\,\left(r\delta_{(0,0)} + (1-r)\,\delta_{(1,0)}\right) + (1-s)\,\delta_{(1/2,1)}$$

with $r,s \in (0,1)$. Substitution in the definition (1) gives $f_{\mathbf{1}_{[1/2,1]},H}\,(\mu_{s,r}) = r\mathbf{1}_{[1/2,1]}\,(r)$.

Now let the underlying law $\mu$ be $\mu = \mu_{1/2,1/2}$. Then $f_{\mathbf{1}_{[1/2,1]},H}\,(\mu) = 1/2$ but the empirical measure is $\hat{\mu} = \mu_{\hat{S},\hat{R}}$, where $\hat{S}$ and $\hat{R}$ can be generated by trials of a fair coin and have binomial distributions concentrated around $1/2$. As the sample size $n$ goes to infinity $\Pr\left\{\hat{R} < 1/2\right\} \to 1/2$ and for every $\epsilon > 0$ we, $\Pr\left\{\left|\hat{R} - 1/2\right| > \epsilon\right\} \to 0$. It follows that

$$\lim_{n \to \infty} \mathbb{E}\left[f_{W,H}\,\left(\mu_{\hat{S},\hat{R}}\right)\right] = \lim_{n \to \infty} \mathbb{E}\left[\hat{R}\,\mathbf{1}_{[1/2,1]}\,\left(\hat{R}\right)\right] = 1/4 < 1/2 = f_{\mathbf{1}_{[1/2,1]},H}\,(\mu)\,.$$

■

## C Proof of Lemma 5

**Lemma 5** *Let $\delta \in (0,1)$ and $A_\delta$ as in (2). Then with probability at least $1 - \delta/2$ it holds that $A_\delta$ implies both $\mu(0) \geq \hat{\mu}(0)/2$ and*

$$\left| f_{W,\ell}(\hat{\mu}) - f_{W,\ell}(\mu) \right| \leq \frac{\left| g_{W,\ell,\mu(0)}(\hat{\mu}) - g_{W,\ell,\mu(0)}(\mu) \right|}{\hat{\mu}(0)\,\hat{\mu}(1)} + \frac{\|W\|_{Lip} + 8\,\|W\|_\infty}{\min\{\hat{\mu}(0),\hat{\mu}(1)\}^2} \sqrt{\frac{2\ln(4/\delta)}{n}}.$$

**Proof.** We have $f_{W,\ell}(\mu) = g_{W,\ell,\mu(0)}(\mu) / (\mu(0)\,\mu(1))$. Thus

$$\left| f_{W,\ell}(\hat{\mu}) - f_{W,\ell}(\mu) \right|$$

$$= \left| \left( \frac{g_{W,\ell,\mu(0)}(\hat{\mu})}{\hat{\mu}(0)\,\hat{\mu}(1)} - \frac{g_{W,\ell,\mu(0)}(\mu)}{\hat{\mu}(0)\,\hat{\mu}(1)} \right) + \left( f_W(\hat{\mu}) - \frac{g_{W,\ell,\mu(0)}(\hat{\mu})}{\hat{\mu}(0)\,\hat{\mu}(1)} \right) \right.$$

$$\left. + \left( \frac{g_{W,\ell,\mu(0)}(\mu)}{\hat{\mu}(0)\,\hat{\mu}(1)} - \frac{g_{W,\ell,\mu(0)}(\mu)}{\mu(0)\,\mu(1)} \right) \right|$$

$$\leq \frac{\left| g_{W,\ell,\mu(0)}(\hat{\mu}) - g_{W,\ell,\mu(0)}(\mu) \right|}{\hat{\mu}(0)\,\hat{\mu}(1)} + \|W\|_{Lip} \left| \frac{1}{\hat{\mu}(0)} - \frac{1}{\mu(0)} \right|$$

$$+ \|W\|_\infty \left| \frac{1}{\hat{\mu}(0)\,\hat{\mu}(1)} - \frac{1}{\mu(0)\,\mu(1)} \right|$$

$$\leq \frac{\left| g_{W,\ell,\mu(0)}(\hat{\mu}) - g_{W,\ell,\mu(0)}(\mu) \right|}{\hat{\mu}(0)\,\hat{\mu}(1)} + \frac{\|W\|_{Lip}}{\hat{\mu}(0)\,\mu(0)} \left| \hat{\mu}(0) - \mu(0) \right|$$

$$+ \left( \frac{\|W\|_\infty}{\hat{\mu}(0)\,\hat{\mu}(1)\,\mu(0)\,\mu(1)} \left| \hat{\mu}(0)\,\hat{\mu}(1) - \mu(0)\,\mu(1) \right| \right)$$

Hoeffdings inequality [5] implies with probability at least $1 - \delta/2$ that

$$\left| \hat{\mu}(1) - \mu(1) \right| = \left| \hat{\mu}(0) - \mu(0) \right| \leq \sqrt{\frac{\ln(4/\delta)}{2n}},$$

so under $A_\delta$ we have $\mu(0) \geq \hat{\mu}(0)/2$ and $\mu(1) \geq \hat{\mu}(1)/2$, and substitution above gives

$$\left| f_{W,\ell}(\hat{\mu}) - f_{W,\ell}(\mu) \right|$$

$$\leq \frac{\left| g_{W,\ell,\mu(0)}(\hat{\mu}) - g_{W,\ell,\mu(0)}(\mu) \right|}{\hat{\mu}(0)\,\hat{\mu}(1)} \left( \frac{\|W\|_{Lip}}{\hat{\mu}(0)^2} + \frac{8\,\|W\|_\infty}{\min\{\hat{\mu}(0),\hat{\mu}(1)\}^2} \right) \sqrt{\frac{2\ln(4/\delta)}{n}}$$

∎

## D Proof of the Lipschitz bound, Proposition 10

For the proof of this proposition we need some preliminary definitions and lemmata. We define

$$K(t) := \int_0^t W\left(\frac{u}{c}\right) du, \quad F_\mu(t) := \mu_0(-\infty, t] \text{ and } \phi_\mu(t) := \int_{\mathbb{R}} \ell(t - t')\, d\mu_1(t'),$$

so $g = g_{W,\ell,c}$ can be written as a Riemann-Stieltjes Integral

$$g_{W,\ell,c}(\mu) = \int_{\mathbb{R}} \phi_\mu(t)\, dK(F_\mu(t)). \tag{5}$$

**Lemma 6** *Let $\mu, \nu, \rho \in \mathcal{P}_1(\mathcal{I} \times \{0,1\})$ and $\ell$ and $W$ be bounded. Then $\left| \phi_\rho(t)\left( K(F_\mu(t)) - K(F_\nu(t)) \right) \right| \to 0$ as $t \to \pm\infty$. Furthermore*

$$\int_{\mathbb{R}} \phi_\rho(t)\left( dK(F_\mu(t)) - dK(F_\nu(t)) \right) = - \int_{\mathbb{R}} \phi'_\rho(t)\left( K(F_\mu(t)) - K(F_\nu(t)) \right). \tag{6}$$

**Proof.**

$$|\phi_\rho(t)\left(K\left(F_\mu(t)\right)-K\left(F_\nu(t)\right)\right)| = \left|\phi_\rho(t)\int_{F_\nu(t)}^{F_\mu(t)}W\left(\frac{u}{c}\right)du\right|$$

$$\leq \|\ell\|_\infty\|W\|_\infty|F_\mu(t)-F_\nu(t)|\to 0$$

as $t\to\pm\infty$. The identity follows from integration by parts. ∎

**Lemma 7** *For $\mu,\mu'\in\mathcal{P}_1\left(\mathcal{I}\times\{0,1\}\right)$*

*(i)* $\|\phi_\mu-\phi_{\mu'}\|_\infty\leq d_\infty\left(\mu_1,\mu_1'\right)$

*If $\ell$ is Lipschitz, then all of the following hold*

*(ii)* $\|\phi_\mu-\phi_{\mu'}\|_\infty\leq\|\ell\|_{Lip}\,d_1\left(\mu_1,\mu_1'\right)$

*(iii)* $\left\|\phi_\mu'-\phi_{\mu'}'\right\|_\infty\leq\|\ell\|_{Lip}\,d_{TV}\left(\mu_1,\mu_1'\right)$

*(iv)* $\left\|\phi_\mu'-\phi_{\mu'}'\right\|_1\leq d_{TV}\left(\mu_1,\mu_1'\right)$

*(v)* $\left\|\phi_\mu'\right\|_\infty\leq\|\ell\|_{Lip}$

*(vi)* $\left\|\phi_\mu'\right\|_1\leq 1$

**Proof.** (i) Recall that $\ell$ is assumed to be non-decreasing with $\ell_n:\mathbb{R}\to[0,1]$. Approximate $\ell$ pointwise from below by non-decreasing, differentiable functions $\ell_n:\mathbb{R}\to[0,1]$, $n\in\mathbb{N}$. By dominated convergence and integration by parts

$$|\phi_\mu(t)-\phi_{\mu'}(t)| = \lim_{n\to\infty}\left|\int_\mathbb{R}\ell_n\left(t'-t\right)d\left(\mu_1-\mu_1'\right)\left(t'\right)\right|$$

$$= \lim_{n\to\infty}\left|\int_\mathbb{R}\ell_n'\left(t'-t\right)\left(F\left(\mu_1,t'\right)-F\left(\mu_1',t'\right)\right)dt'\right|$$

$$\leq d_\infty\left(\mu,\mu'\right)\lim_{n\to\infty}\int_\mathbb{R}\left|\ell_n'\left(t'-t\right)\right|dt' = d_\infty\left(\mu,\mu'\right)\lim_{n\to\infty}\int_\mathbb{R}\ell_n'\left(t'-t\right)dt'$$

$$\leq d_\infty\left(\mu,\mu'\right).$$

by the assumed properties of $\ell$.

If $\ell$ is Lipschitz, then $\ell$ is absolutely continuous and its derivative $\ell'$ exists almost everywhere and $\|\ell\|_{Lip}=\|\ell'\|_\infty$. (ii)

$$|\phi_\mu(t)-\phi_{\mu'}(t)| \leq \left|\int_\mathbb{R}\ell\left(t'-t\right)d\left(\mu_1-\mu_1'\right)\left(t'\right)\right|$$

$$= \left|\int_\mathbb{R}\ell'\left(t'-t\right)\left(F\left(\mu_1,t'\right)-F\left(\mu_1',t'\right)\right)dt'\right|$$

$$\leq \|\ell'\|_\infty\,d_1\left(\mu_1,\mu_1'\right)$$

(iii)

$$\left|\phi_\mu'(t)-\phi_{\mu'}'(t)\right| = \left|\int_\mathbb{R}\ell'\left(t'-t\right)d\left(\mu_1-\mu_1'\right)\left(t'\right)\right|\leq\|\ell'\|_\infty\,d_{TV}\left(\mu_1,\mu_1'\right)$$

(iv)

$$\int\left|\phi_\mu'(t)-\phi_{\mu'}'(t)\right|dt = \int_\mathbb{R}\left|\int_\mathbb{R}\ell'\left(t'-t\right)d\left(\mu_1-\mu_1'\right)\left(t'\right)\right|dt$$

$$\leq \int_\mathbb{R}\int_\mathbb{R}\left|\ell'\left(t'-t\right)\right|d\left(|\mu_1-\mu_1'|\right)\left(t'\right)dt$$

$$= \int_\mathbb{R}\left(\int_\mathbb{R}\ell'\left(t-t'\right)dt\right)d\left(|\mu_1-\mu_1'|\right)\left(t'\right)$$

$$= \int_\mathbb{R}d\left(|\mu_1-\mu_1'|\right)\left(t'\right) = d_{TV}\left(\mu_1,\mu_1'\right).$$

The proofs of (v) and (vi) are similar to those of (iii) and (iv). ∎

Now we can prove Proposition 10.

**Proposition 10** *Fix $W, \ell$ and $c$. The functional $g = g_{W,\ell,c}$ satisfies $\forall \mu, \mu', \nu, \nu' \in P_1(\mathcal{I} \times \{0,1\})$*

*(a) $g(\nu) - g(\nu') \le \|W\|_\infty (d_\infty(\nu_0, \nu'_0) + d_\infty(\nu_1, \nu'_1))$*

*(b) if $\ell$ is Lipschitz then*

$$g(\nu) - g(\nu') \le \|W\|_\infty \|\ell\|_{Lip} (d_1(\nu_0, \nu'_0) + d_1(\nu_1, \nu'_1))$$

*(c) if $W$ and $\ell$ are Lipschitz then*

$$g(\lambda(\mu, \nu)) - g(\lambda(\mu, \nu')) - g(\lambda(\mu', \nu)) + g(\lambda(\mu', \nu'))$$
$$\le (1 - \lambda) \lambda \|\ell\|_{Lip} \left( \frac{1}{c} \|W\|_{Lip} + \|W\|_\infty \right) (d_1(v_1, v'_1) + d_1(\nu_0, \nu'_0)) d_{TV}(\mu, \mu').$$

**Proof.** From (5) we have

$$g(\mu) - g(\nu) = \int_{\mathbb{R}} (\phi_\mu(t) - \phi_\nu(t)) \, dK(F_\mu(t)) + \int_{\mathbb{R}} \phi_{\nu,n}(t) (dK(F_\mu(t)) - dK(F_\nu(t))).$$

Proof of (a). Approximate $\ell$ as in the proof of Lemma 7 (i) and let

$$\phi_{\nu,n}(t) = \int_{\mathbb{R}} \ell_n(t - t') \, d\mu_1(t').$$

Using dominated convergence, the integration by parts formula (6) and Lemma 7 (i)

$$g(\mu) - g(\nu)$$
$$= \int_{\mathbb{R}} (\phi_\mu(t) - \phi_\nu(t)) W\left(\frac{F_\mu(t)}{c}\right) d\mu_0(t) - \lim_{n \to \infty} \int_{\mathbb{R}} \phi'_{\nu,n}(t) (K(F_\mu(t)) - K(F_\nu(t))) \, dt$$
$$\le \|W\|_\infty \left( \|\phi_\mu(.) - \phi_\nu(.)\|_\infty + \lim_{n \to \infty} \int_{\mathbb{R}} \phi'_{\nu,n}(t) |F_\mu(t) - F_\nu(t)| \, dt \right)$$
$$\le \|W\|_\infty \left( d_\infty(\mu_1, \nu_1) + d_\infty(\mu_0, \nu_0) \lim_{n \to \infty} \int_{\mathbb{R}} \phi'_{\nu,n}(t) \, dt \right)$$
$$\le \|W\|_\infty (d_\infty(\mu_1, \nu_1) + d_\infty(\mu_0, \nu_0)).$$

Proof of (b). Now $\ell$ is Lipschitz and we can write similar to the above

$$g(\mu) - g(\nu) \le \|W\|_\infty \left( \|\phi_\mu(.) - \phi_\nu(.)\|_\infty + \int_{\mathbb{R}} \phi'_\nu(t) |F_\mu(t) - F_\nu(t)| \, dt \right)$$
$$\le \|W\|_\infty \|\ell\|_{Lip} d_1(\mu_1, \nu_1) + d_1(\mu_0, \nu_0),$$

where we used Lemma 7 (ii) and (v).

Proof of (c). We write the second difference as

$$g(\lambda(\mu, \nu)) - g(\lambda(\mu, \nu')) - g(\lambda(\mu', \nu)) - g(\lambda(\mu', \nu'))$$
$$= \int_{\mathbb{R}} \left( \phi_{\lambda(\mu,\nu)}(t) - \phi_{\lambda(\mu,\nu')}(t) - \phi_{\lambda(\mu',\nu)}(t) + \phi_{\lambda(\mu',\nu')}(t) \right) dK\left(F_{\lambda(\mu,\nu)}(t)\right)$$
$$+ \int_{\mathbb{R}} \left( \phi_{\lambda(\mu',\nu)}(t) - \phi_{\lambda(\mu',\nu')}(t) \right) \left( dK\left(F_{\lambda(\mu,\nu)}(t)\right) - dK\left(F_{\lambda(\mu',\nu)}(t)\right) \right)$$
$$+ \int_{\mathbb{R}} \left( \phi_{\lambda(\mu,\nu')}(t) - \phi_{\lambda(\mu',\nu')}(t) \right) \left( dK\left(F_{\lambda(\mu,\nu)}(t)\right) - dK\left(F_{\lambda(\mu,\nu')}(t)\right) \right)$$
$$+ \int_{\mathbb{R}} \phi_{\lambda(\mu',\nu')}(t) \left( dK\left(F_{\lambda(\mu,\nu)}(t)\right) - dK\left(F_{\lambda(\mu,\nu')}(t)\right) - dK\left(F_{\lambda(\mu',\nu)}(t)\right) + dK\left(F_{\lambda(\mu',\nu')}(t)\right) \right)$$
$$= A + B + C + D.$$

We bound the four terms in turn. The term $A$ simply vanishes, because $\phi_\mu$ is affine in $\mu$. To bound $B$ first note that for any $t$

$$K\left(F_{\lambda(\mu',\nu)}(t)\right) - K\left(F_{\lambda(\mu,\nu)}(t)\right) = \int_{F_{\lambda(\mu,\nu)}(t)}^{F_{\lambda(\mu',\nu)}(t)} W\left(\frac{u}{c}\right) du = \int_{(1-\lambda)F_\mu(t)}^{(1-\lambda)F_{\mu'}(t)} W\left(\frac{u}{c} - \lambda F_\nu(t)\right) du$$

$$\leq (1-\lambda)\|W\|_\infty |F_{\mu'}(t) - F_\mu(t)|.$$

Using integration by parts and Lemma 7 (iii)

$$B = \int_{\mathbb{R}} \left(\phi'_{\lambda(\mu',\nu)}(t) - \phi'_{\lambda(\mu',\nu')}(t)\right)\left(K\left(F_{\lambda(\mu',\nu)}(t)\right) - K\left(F_{\lambda(\mu,\nu)}(t)\right)\right) dt$$

$$\leq (1-\lambda)\|W\|_\infty \int_{\mathbb{R}} \left|\phi'_{\lambda(\mu',\nu)}(t) - \phi'_{\lambda(\mu',\nu')}(t)\right| |F_{\mu'}(t) - F_\mu(t)| \, dt$$

$$\leq (1-\lambda)\lambda \|\ell'\|_\infty \|W\|_\infty \, d_1(\mu_0,\mu'_0) \, d_{TV}(\nu_1,\nu'_1)$$

The term $C$ is bounded using Lemma 7 (ii) as

$$C = \int_{\mathbb{R}} \left(\phi_{\lambda(\mu,\nu')}(t) - \phi_{\lambda(\mu',\nu')}(t)\right)\left(dK\left(F_{\lambda(\mu,\nu)}(t)\right) - dK\left(F_{\lambda(\mu,\nu')}(t)\right)\right)$$

$$\leq \left\|\phi_{\lambda(\mu,\nu')}(.) - \phi_{\lambda(\mu',\nu')}(.)\right\|_\infty \left(\int_{\mathbb{R}} \left|W\left(\frac{F_{\lambda(\mu,\nu)}(t)}{c}\right) - W\left(\frac{F_{\lambda(\mu,\nu')}(t)}{c}\right)\right| d\mu_0(t)\right.$$

$$\left. + \int_{\mathbb{R}} W\left(\frac{F_{\lambda(\mu,\nu')}(t)}{c}\right) |d\nu_0(t) - d\nu'_0(t)|\right)$$

$$\leq (1-\lambda)\lambda \|\ell'\|_\infty \left(\left\|\frac{W}{c}\right\|_{Lip} + \|W\|_\infty\right) d_1(\mu_1,\mu'_1) \, d_{TV}(\nu_0,\nu'_0).$$

Finally we again use integration by parts to bound $D$.

$$D = \int_{\mathbb{R}} \phi'_{\lambda(\mu',\nu')}(t)\left(K\left(F_{\lambda(\mu,\nu)}(t)\right) - K\left(F_{\lambda(\mu,\nu')}(t)\right) - K\left(F_{\lambda(\mu',\nu)}(t)\right) + K\left(F_{\lambda(\mu',\nu')}(t)\right)\right) dt$$

$$= \int_{\mathbb{R}} \phi'_{\lambda(\mu',\nu')}(t)\left(\int_{F_{\lambda(\mu,\nu')}(t)}^{F_{\lambda(\mu,\nu)}(t)} W\left(\frac{u}{c}\right) du - \int_{F_{\lambda(\mu',\nu')}(t)}^{F_{\lambda(\mu',\nu)}(t)} W\left(\frac{u}{c}\right) du\right) dt$$

$$= \int_{\mathbb{R}} \phi'_{\lambda(\mu',\nu')}(t)\left(\int_{F_{\lambda\nu'}(t)}^{F_{\lambda\nu}(t)} \left(W\left(\frac{u-(1-\lambda)F_\mu(t)}{c}\right) - W\left(\frac{u-(1-\lambda)F_{\mu'}(t)}{c}\right)\right) du\right) dt$$

$$\leq (1-\lambda)\lambda \left\|\frac{W}{c}\right\|_{Lip} \int_{\mathbb{R}} \left|\phi'_{\lambda(\mu',\nu')}(t)\right| |F_\nu(t) - F_{\nu'}(t)| \, |F_\mu(t) - F_{\mu'}(t)| \, dt$$

$$\leq (1-\lambda)\lambda \|\ell'\|_\infty \left\|\frac{W}{c}\right\|_{Lip} \int_{\mathbb{R}} |F_\nu(t) - F_{\nu'}(t)| \, |F_\mu(t) - F_{\mu'}(t)| \, dt$$

$$\leq (1-\lambda)\lambda \|\ell'\|_\infty \left\|\frac{W}{c}\right\|_{Lip} d_1(\mu_0,\mu'_0) \, d_{TV}(\nu_0,\nu'_0),$$

where we used Lemma 7 (v) in the second inequality. Adding the bounds and using $d_{TV}(\nu_i,\nu'_i) \leq d_{TV}(\nu,\nu')$, we get

$$A + B + C + D \leq (1-\lambda)\lambda \|\ell'\|_\infty \left(\left\|\frac{W}{c}\right\|_{Lip} + \|W\|_\infty\right)(d_1(\mu_0,\mu'_0) + d_1(\mu_1,\mu'_1)) \, d_{TV}(\nu,\nu')$$

∎

## E  Proof of Lemma 11

**Lemma 11** *] For $\mu,\nu \in P_1(\mathcal{I} \times \{0,1\})$ we have*

*(i)* $d_{TV}(\mu_0,\nu_0) + d_{TV}(\mu_1,\nu_1) = d_{TV}(\mu,\nu) = d_{TV}(\tau_\#\mu, \tau_\#\nu)$

*(ii)* $d_\infty(\mu_0,\nu_0) + d_\infty(\mu_1,\nu_1) \leq 2d_\infty(\tau_\#\mu, \tau_\#\nu)$

*(iii)* $d_1(\mu_0,\nu_0) + d_1(\mu_1,\nu_1) \leq 2d_1(\tau_\#\mu, \tau_\#\nu)$

**Proof.** (i) is obvious from the definitions of $\mu_i, \nu_i$ and $\tau_{\#}\mu, \tau_{\#}\nu$.

(ii)

$$
\begin{aligned}
&d_\infty\left(\mu_0, \nu_0\right) + d_\infty\left(\mu_1, \nu_1\right) \\
&= \sup_{t \in \mathcal{I}} \left|\mu_0\left(-\infty, t\right] - \nu_0\left(-\infty, t\right]\right| + \sup_{t \in \mathcal{I}} \left|\mu_1\left(-\infty, t\right] - \nu_1\left(-\infty, t\right]\right| \\
&= \sup_{t \in \mathcal{I}} \left|\tau_{\#}\mu\left(-\infty, t-a\right] - \tau_{\#}\nu\left(-\infty, t-a\right]\right| + \sup_{t \in \mathcal{I}} \left|\tau_{\#}\mu\left[a-t, \infty\right) - \tau_{\#}\nu\left[a-t, \infty\right)\right| \\
&\leq \sup_{t \in \mathbb{R}} \left|\tau_{\#}\mu\left(-\infty, t\right] - \tau_{\#}\nu\left(-\infty, t\right]\right| + \sup_{t \in \mathbb{R}} \left|\tau_{\#}\mu\left[t, \infty\right) - \tau_{\#}\nu\left[t, \infty\right)\right| \\
&\leq 2 d_\infty\left(\tau_{\#}\mu, \tau_{\#}\nu\right).
\end{aligned}
$$

(iii)

$$
\begin{aligned}
&2 d_1\left(\tau_{\#}\mu, \tau_{\#}\nu\right) \\
&= \int_{-\infty}^{\infty} \left|\tau_{\#}\mu\left(-\infty, t\right] - \tau_{\#}\nu\left(-\infty, t\right]\right| dt + \int_{-\infty}^{\infty} \left|\tau_{\#}\mu\left[t, \infty\right) - \tau_{\#}\nu\left[t, \infty\right)\right| dt \\
&\geq \int_{-\infty}^{0} \left|\tau_{\#}\mu\left(-\infty, t\right] - \tau_{\#}\nu\left(-\infty, t\right]\right| dt + \int_{0}^{\infty} \left|\tau_{\#}\mu\left[t, \infty\right) - \tau_{\#}\nu\left[t, \infty\right)\right| dt \\
&= \int_{\mathcal{I}} \left|\tau_{\#}\mu\left(-\infty, t-a\right] - \tau_{\#}\nu\left(-\infty, t-a\right]\right| dt + \int_{\mathcal{I}} \left|\tau_{\#}\mu\left[a-t, \infty\right) - \tau_{\#}\nu\left[a-t, \infty\right)\right| dt \\
&= \int_{\mathcal{I}} \left|\left|\mu_0\left(-\infty, t\right] - \nu_0\left(-\infty, t\right]\right|\right| dt + \int_{\mathcal{I}} \left|\mu_1\left(-\infty, t\right] - \nu_1\left(-\infty, t\right]\right| dt \\
&= d_1\left(\mu_0, \nu_0\right) + d_1\left(\mu_1, \nu_1\right)
\end{aligned}
$$

∎

# F Table of notation

| Notation | Definition | Section |
|---|---|---|
| $A_\delta$ | Conditioning event | 2.1 |
| $d_{TV}(\mu,\nu)$ | Total variation distance for $\mu,\nu \in \mathcal{P}_1(\mathcal{X})$ | 3.2 |
| $d_1(\mu,\nu)$ | 1-Wasserstein distance for $\mu,\nu \in \mathcal{P}_1(\mathbb{R})$ | 3.2 |
| $d_\infty(\mu,\nu)$ | Kolmogorov distance for $\mu,\nu \in \mathcal{P}_1(\mathbb{R})$ | 3.2 |
| $D_{y,y'}^k$ | Partial difference operator | 3.2 |
| $\delta_x$ | Unit mass at $x$ | 1.2 |
| $fpr$ | False positive rate | 2 |
| $f_\#\mu$ | Pushforward of measure $\mu$ under $f$ | 1.2 |
| $f_{W,\ell}$ | Weighted area und ROC curve | 2 |
| $g_{W,\ell,c}$ | Functional independent of label frequancies | 3.1 |
| $G(\mathcal{H})$ | Gaussian complexity | 2.2 |
| $H$ | Heaviside function $1_{(0,\infty)}$ | 1.2 |
| $\mathcal{H}$ | Class of scoring functions | 2.2 |
| $\bar{h}$ | $\bar{h}(x,y) = (h(x),y)$ for $h \in \bar{h}$ | 2.2 |
| $\ell$ | Loss function | 2 |
| $\mu_0$ | $\mu_0(A) = \mu(A \times \{0\})$ for $\mu \in \mathcal{P}_1(\mathcal{X} \times \{0,1\})$ | 1.2 |
| $\mu_1$ | $\mu_1(A) = \mu(A \times \{1\})$ for $\mu \in \mathcal{P}_1(\mathcal{X} \times \{0,1\})$ | 1.2 |
| $\hat{\mu}(\mathbf{X})$ | Empirical measure for sample $\mathbf{X}$ | 1.2 |
| $\mathcal{P}(\mathcal{X})$ | Nonnegative measures on $\mathcal{X}$ | 1.2 |
| $\mathcal{P}_1(\mathcal{X})$ | Probability measures on $\mathcal{X}$ | 1.2 |
| $roc$ | ROC-curve | 2 |
| $tpr$ | True positive rate | 2 |
| $\tau$ | Bijection $I \times \{0,1\} \to (-b,0) \cup (0,b)$ | 3.3 |
| $W$ | Weight function | 2 |
| $\|\cdot\|_\infty$ | Supremum norm | 1.2 |
| $\|\cdot\|_{Lip}$ | Lipschitz seminorm | 1.2 |