[Reviews · NeurIPS 2020]

Review 1

Summary and Contributions: The paper considers generalization bounds for weighted area under the ROC curve for Lipschitz weight functions. The paper first shows that if the weight function is non-Lipschitz e.g. discontinuous, then there exist adversarially choosen distributions such that consistent estimation is impossible even in the limit. The paper then offers generalization bounds for fixed scoring functions, as well as uniform convergence bounds to address cases where the scoring function is itself learnt.

Strengths: Soundness of the claims (theoretical grounding, empirical evaluation) -------------------------------- The guarantees are sound and seem tight (but for log factors). The explicit lower bound in Proposition 1 is interesting and offers a handy example for an otherwise folklore belief.

Weaknesses: Soundness of the claims (theoretical grounding, empirical evaluation) -------------------------------- Most of the discussion focusses on Lipschitz weight functions and scant attention is paid to the more interesting case of non-Lipschitz weight functions. Significance and novelty of the contribution -------------------------------- The paper does not clarify what are the specific challenges faced to acheive the results. One contribution seems to have been in defining a surrogate functional g (line 166) that replaces the \mu(0) term in a denominator term with an arbitrary parameter c and then using a uniform convergence bound over values of c to ensure that estimation does take place even if c is replaced with its actual value of \mu(0). Another contribution seems to be in fine tuning the proof technique used to prove Proposition 5. However, it is not clear how much of the heavy lifting was already taken care of by [17] when it established Theorem 8. Relevance to the NeurIPS community -------------------------------- The paper offers no algorithmic intuitions, nor any explicit algorithms. The main contribution is a proof for obtaining generalization bound for weighted areas under the ROC curve for Lipschitz weight functions.

Correctness: The proposition in line 106-108 is not entirely convincing. Partial AUC is a very well studied problem [19, 20, 21] and indeed corresponds to a discontinuous weight function W. The way around is to indeed assume that the underlying law \mu does not involve atomic masses, which is reasonable in most practical situations.

Clarity: The paper is notation heavy which may be intimidating to some readers. More importantly, discussions on the most interesting case of discontinuous weight functions is relegated to remarks which briefly outline how the results which are proved for Lipschitz weight functions can be extended to discontinuous ones. Please consider devoting properly stated ways to use Theorem 2, 3 and "sandwiching" results (e.g. sandwich a weight function between two others) to address discontinuous weight functions. At a couple of places, notation is used without proper introduction (please see below).

Relation to Prior Work: There is prior work (see Theorem 4 in the citation below) that effectively offers generalization bounds for discontinuous partial area under the ROC curve by offering an online to batch conversion bound. Since this work directly gives a generalization bound for a non-Lipschitz weight function case instead of requiring sandwiching etc, it is relevant to the paper. Kar, Narasimhan and Jain, Online and Stochastic Gradient Methods for Non-decomposable Loss Functions, NIPS 2014.

Reproducibility: Yes

Additional Feedback: line number 46: Heavyside => Heaviside (wrong name) The notation in the paper is a bit confusing. The notation X is usually reserved for feature vectors or covariates but it seems that in the paper it is abused to denote scores (e.g. on line 59) that some scoring function has provided. However, line 94 suggests that it is still being used to describe "data" i.e. features. This causes confusion e.g. on line 130 where it is unclear to what does X refer, a score or a feature vector. Can you please clarify? Please explain the notation Z_i on line 97. It seems to have been used without prior introduction or definition. Remark 3 on line 119 follows remark 1 i.e. remark 2 is missing. Please explain the notation F in the equation after line 166. It seems it has not been introduced or defined anywhere.


Review 2

Summary and Contributions: The paper regards estimation of weighted AUC of a scoring function to take into account differential costs of false positives and false negatives. The authors provide finite sample bounds for the estimation error of an empirical partial AUC (pAUC) as a plug-in estimator of the population version under the assumption that the weight function is Lipschitz. Further they establish a uniform estimation error bound over a class of scoring functions using the Gaussian complexity measure, providing a theoretical guarantee for empirical pAUC maximization.

Strengths: The paper provides some theoretical justifications for ranking algorithms that are designed to optimize the sample pAUC. The results extend the existing theoretical analyses for AUC optimization, and I consider them fine and solid theoretical contributions. It was interesting to see the challenge in estimation of pAUC with a discontinuous weight function. Implications of the results for specifying a weight function would be of practical importance although they are not clearly stated in the paper.

Weaknesses: What I find missing in the paper is a broad perspective bridging existing results and the new contributions. The authors mention several papers on asymptotic results for the pAUC estimator and its connection to L-estimators in classical statistics literature. Finite sample bounds also have implications for asymptotics. In addition, asymptotic analysis can provide fairly good finite sample approximations as well. I suspect that the generalization error bounds for pAUC in the paper might be quite loose as in many worst-case analyses. Critical comparisons with alternative approaches and more concrete examples for illustration of the theoretical results will definitely broaden the scope of the work. -- Post Rebuttal Update -- The author feedback didn't change much of my opinion on the weakness of the paper. It partially addresses my comment on Section 1.1 related work, but I didn't find the response adds sufficiently different information or insightful perspective.

Correctness: Questions for clarification: In Theorem 3 and Proposition 5, the distribution for (X,Y) must be \mu^n rather than \mu hat. Line 97: What is Z_i? Was it ever defined? Line 136: Isn’t \ell Lipschitz lower bound of H rather than \ell itself? Between line 166 and 167: What is F(,) function? Is it cdf? A minor typo on line 55: “ared defined”

Clarity: Due to the very technical nature of the work, the paper is quite dense. Pretty much all math and no examples, illustration or application. Nevertheless, it is overall well written in technical terms to the extent that I could understand.

Relation to Prior Work: Section 1.1 discusses related previous contributions. While the authors are successful in saying that the presented results are genuinely new and different from the previous ones, I find very little perspectives in the section as to how they are connected. For example, there is a mention of finite sample bounds for specific distributions in some earlier work. What is the relation between the general bounds in the paper and the more specific bounds? Does the difference imply any additional insight or room for sharpening the general bounds?

Reproducibility: Yes

Additional Feedback:


Review 3

Summary and Contributions: Paper considered the theoretical property of AUC plugin estimator. Error bound and uniform error bound is established. -- Thanks for the response from the authors. I am changing the score to 6. I am still hoping the authors including more algorithm/simulation/application associated contents in the final draft and could move the proof details to appendix to address the page limit.

Strengths: The bounds are carefully developed and the math foundation of AUC is studied. The bounds are useful for theoretical ML work and algorithm analysis.

Weaknesses: This is an interesting theoretical paper dealing with some properties of a statistic that is very widely used. However I am having difficulties selling this paper to the wider audience, as AUC users are not quite interested in studying the theoretical properties. Also lacking numerical simulations would be a potential room for improvement of the paper

Correctness: I did not check the proof in detail but the bounds and the uniform bounds are of good mathmatical interest and the form looks solid.

Clarity: Yes

Relation to Prior Work: AUC works are listed. Not sure about the work on the theoretical analysis on AUC. Might because it is limited in quantity.

Reproducibility: Yes

Additional Feedback:


Review 4

Summary and Contributions: The area under the ROC curve (AUC) has long been used to evaluate the scoring function. In some cases, however, the different regions of the range of false positives may not be as relevant for assessing the quality of the score and therefore need to be weighted differently. This is referred to as pAUC for partial area under the ROC curve. Several algorithms have been designed to maximize both AUC and pAUC. The AUC framework has been studied extensively with efficient results. However, regarding the pAUC setting, the estimation may be difficult to impossible. The contributions are twofold. First, the authors prove that the weighted AUC can be efficiently estimated, as soon as the weight is assumed to be Lipschitz continuous. They also characterize the speed of convergence as a function of sample size. Then, the authors provide a general method to control the estimation error of any functional of the same type as the weighted AUC, under Lipschitz conditions.

Strengths: The article is well written and comprehensive. The authors have taken care to define their study framework before embarking on the demonstrations. No evidence is omitted; they are accurately written and accessible. This paper provides theoretical guaranties to some algorithms used in practice (especially for medical diagnostics).

Weaknesses: Some passages are quite difficult to understand. For example, I had to reread several times the remark 3 page 4 to try to grasp what the authors meant (and in the end I'm not 200% sure of myself!). There are a lot of notations, which requires a lot of back and forth throughout the reading. Perhaps adding an appendix section that summarizes the definition of f_{W,l), g, etc. would help? Concretely, I have had several times the urge to reread a definition and I wasted several minutes looking for it because important definitions are scattered all over the paper (in fact, where it makes sense to introduce them). Moreover, the writing in free text does not really bring them out; it was sometimes tedious (without being prohibitive).

Correctness: Unless I am mistaken, the proofs are correct. I just picked up a few typos: (i) There is no remark 2 page 4. (ii) In the statement of Theorem 6, line 194, I think that two v should be \nu. (iii) In the proof of Theorem 8, line 219, there is an apostrophe missing from the last y (the step with with f()-f()-f()+f(….y’) < ... ). (iv) Same line, there is a L_12 instead of a L_2 in the first inequality. (v) In the appendix, proof of Lemma 4, line 344, a l is missing in a f_{W,l} (second line of the calculus). (vi) Proof of Proposition 9, line 359 and 360. I think that it should be a \gamma(\lambda,s) instead of a \gamma(0,s). Many derivatives/integral inversions seem to be made without taking into account the regularity of the functions involved.

Clarity: Good readability and organization. A few passages could be clarified: (i) The first sentence of paragraph 1.2 sounds strange. (ii) The remark 3 page 4 seems obscure to me. (iii) The enumeration of the proposition in the appendix is not consistent with the one of the main paper. (iv) For the appendices, prefer a numbering of the sections based on letters. (v) Write the equations of line 367 on two lines. (vi) In the proof of Proposition 9 (Section 5.4), the arrival of lemmas 15 and 16 seems to me to be abrupt. As well as the return to the demonstration of the proposition after the two small demonstrations (line 384). I propose either to add a few words to explain that one pauses in the main demonstration in order to enunciate/demonstrate two auxiliary lemmas (+ afterward that the main proof is back), or to take the two lemmas out of the proof and to consider them as preliminary results. (vii) Proof of Proposition 9, line 388, references are broken.

Relation to Prior Work: Section 1.1 cites articles but does not really explain the content of the various works. Contextualization could be improved. For example, line 23 the authors state the estimation could be impossible in the pAUC framework, but without bringing support to their assessment.

Reproducibility: Yes

Additional Feedback: Post-Rebuttal : Thanks to the authors for their response. I continue to find this article well-executed and pleasant to follow. So, I maintain a favorable score. Like the other reviewers, I think that this work would be more beneficial to the community if it was accompanied by experiments/illustrations.

[Author Response · NeurIPS 2020]

**Reviewers 1-4**

Many thanks to the four reviewers for their patience with the paper, the typos and inconsistencies they identified and the many suggestions provided. We will fix all the typos you pointed out. For example: line 97: Z_i will be deleted, line 119: renumber and clarify remark, line 136: the last $\ell$ should be $H$, line 166-167: $F(\mu_0, t)$ should be replaced by $\mu_0((-\infty, t])$, et cetera.

**Reviewer 1**

**Most of the discussion focusses on Lipschitz weight functions and scant attention is paid to the more interesting case of non-Lipschitz weight functions.** As shown in Proposition 1 the pAUC with non-Lipschitz weight function cannot be estimated in general, unless nonatomic measures are assumed. Non-Lipschitz weight functions are handled by the bracketing technique (see Remark 3 on page 5 and the proposed corollary below).

**...it is not clear how much of the heavy lifting was already taken care of by [17]**. [17] bounds the difference between the empirical functional and its expectation but not the bias (difference between the expectation and the functional evaluated on the underlying law).

**Partial AUC is a very well studied problem [19, 20, 21] and indeed corresponds to a discontinuous weight function W. The way around is to indeed assume that the underlying law \mu does not involve atomic masses....** None of [19, 20, 21] make this assumption, nor would they need it, because they do not address the problem of estimation.

**...which is reasonable in most practical situations.** Assuming non-atomic distributions of score functions is a severe restriction and rules out discrete features. For example think of a score function depending on whether a patient has a cough or not.

**The paper offers no algorithmic intuitions, nor any explicit algorithms.** [19, 20, 21] all provide elegant algorithms to optimize the empirical error, but they give no generalization guarantees.

**Please consider devoting properly stated ways to use Theorem 2, 3 and "sandwiching" results ... to address discontinuous weight functions.** We will add the following corollary, which should help to clarify Remark 3 on page 5 (with a similar corollary after Theorem 2):

***Corollary*** *(of Theorem 3) Let* $\hat{W}, W_{Lip}, W : [0,1] \to [0,\infty)$, $\hat{W} \le W_{Lip} \le W$ *and* $\|W_{Lip}\|_{Lip} < \infty$. *Then with probability at least* $1 - \delta$ *in* $(\mathbf{X}, \mathbf{Y}) \sim \mu^n$ *we have that* $A_\delta$ *implies*

$$\forall h \in \mathcal{H}, f_{W,H}\left((\bar{h}_{\#}\mu)\right) \ge f_{\hat{W}, \ell}\left(\bar{h}_{\#}\hat{\mu}\right) - B\left(n, W_{Lip}, \mathcal{H}, \hat{\mu}, \delta\right),$$

*where* $B\left(n, W_{Lip}, \mathcal{H}, \hat{\mu}, \delta\right)$ *is the bound in Theorem 3.*

Then $W$ would be the (discontinuous) application window and $\hat{W}$ the (discontinuous) training window.

**There is prior work...** Many thanks for the reference. We will compare to Theorem 4 therein from the perspectives of rates ($n^{-1/4}$ vs our $n^{-1/2}$), dependence on dimensionality, constants and underlying assumptions.

**Notation.** $(X, Y)$ is standard for random labeled instances. We will highlight the distinction between the output of a fixed score function and features.

**Reviewer 2**

**The authors mention several papers on asymptotic results.** The asymptotic results for L-estimators are not directly applicable to the pAUC. We could give Berry-Esseen-type bounds relying on the Lipschitz properties of the pAUC we identified, but we felt this would go beyond the scope of the paper.

**For example, there is a mention of finite sample bounds for specific distributions in some earlier work.** The bounds in [3] are for fixed score functions and special distributions, not so relevant to ML where one wants to select the score function and the distribution is unknown.

**Reviewer 3**

**AUC users are not quite interested in studying the theoretical properties.** Should the theoretical evidence we provide be disqualified by the lack of numerical experiments? Practitioners could well be interested in the conditions under which their algorithms are protected from overfitting.

**Reviewer 4**

**For example, line 23 the authors state the estimation could be impossible in the pAUC framework, but without bringing support to their assessment.** There should be a reference to Proposition 1 in the sentence following line 23.

**I had to reread several times the remark 3 page 4.** We will fix this. Consider also the corollary proposed above in response to Reviewer 1.

**Perhaps adding an appendix section that summarizes the definition of f_{W,l), g, etc. would help?** Adding a table of notation in the appendix is a very good idea.

Special thanks for reading the proofs in the supplement and detailing the various typos there. We will reorganize the sequence of lemmas in the proof of Proposition 9 and enlarge the remark at the beginning of the proof to clarify the regularity assumptions.

**Section 1.1 cites articles but does not really explain the content of the various works.** We will sketch the contents.

[Meta-Review · NeurIPS 2020]

This is a theoretical paper that has received relatively good reviews. However, two of the reviewers only increased their scores from 5 to 6 in order to reduce the divergence and help form a consensus (in the discussions), but neither was really convinced about the quality of the paper. Unfortunately, the highest scoring reviewer was also the least confident. I read the paper myself and I find that it has some merits --- it seems theoretically solid, but I have a slight tendency towards saying that it may be a better fit at ALT/AISTATS/COLT, and it is unclear if the NeurIPS community will benefit from knowing these results. Nevertheless, regardless of the final outcome, the authors are encouraged to improve the readability of their paper through (it is currently somewhat dense for the average reader).